# Eleven Years of Health Monitoring in Wild Boars (*Sus scrofa*) in the Emilia-Romagna Region (Italy)

**DOI:** 10.3390/ani13111832

**Published:** 2023-05-31

**Authors:** Arianna Rossi, Annalisa Santi, Filippo Barsi, Gabriele Casadei, Alessandra Di Donato, Maria Cristina Fontana, Giorgio Galletti, Chiara Anna Garbarino, Annalisa Lombardini, Carmela Musto, Alice Prosperi, Giovanni Pupillo, Gianluca Rugna, Marco Tamba

**Affiliations:** 1Istituto Zooprofilattico Sperimentale della Lombardia e dell’Emilia-Romagna “Bruno Ubertini”, 25124 Brescia, Italyfilippo.barsi@izsler.it (F.B.); gabriele.casadei@izsler.it (G.C.); alessandra.didonato@izsler.it (A.D.D.); mariacristina.fontana@izsler.it (M.C.F.); giorgio.galletti@izsler.it (G.G.); chiaraanna.garbarino@izsler.it (C.A.G.); alice.prosperi@izsler.it (A.P.); giovanni.pupillo@izsler.it (G.P.); gianluca.rugna@izsler.it (G.R.); marco.tamba@izsler.it (M.T.); 2Settore Prevenzione Collettiva e Sanità Pubblica, Direzione Generale Cura della Persona, Salute e Welfare, Emilia-Romagna Region, 40127 Bologna, Italy; annalisalombardini@outlook.it; 3Department of Veterinary Medical Sciences, University of Bologna, 40064 Bologna, Italy; carmela.musto2@unibo.it

**Keywords:** african swine fever, Aujeszky’s disease, brucellosis, classical swine fever, passive surveillance, swine influenza, swine vesicular disease, tuberculosis, trichinellosis, wild boar

## Abstract

**Simple Summary:**

Wildlife monitoring plans are not uniformly applied in all European countries, even if the World Organization for Animal Health (WOAH) identified passive surveillance of wildlife as the strategy of choice to investigate the health status of wild animals. The plan in use in the Emilia-Romagna region (Northern Italy) for the past 11 years has allowed for the collection of a large amount of data on the wild boar population. Research has been conducted on diseases for which the wild boar could be a reservoir and/or source of infection for domestic pigs due to their increasingly frequent interfaces (trichinellosis, tuberculosis, brucellosis, african swine fever, classical swine fever, Aujeszky’s disease, swine vesicular disease, and swine influenza A). Although the results do not allow us to make inferences about the resident population due to the sampling method and sample size, they still give us some indications about the strengths and weaknesses of the plan itself. For instance, an active search for carcasses on the territory should be implemented. In order to improve surveillance activities, it would also be desirable to increase the harmonization of sample collection schemes and data organization from a One Health perspective, as recommended by the WOAH.

**Abstract:**

In recent years, the growth of wild ungulates has increased the focus on their health monitoring. In particular, the health status of wild boars is relevant for the economic impact on the pig industry. The Emilia-Romagna region activated a wildlife monitoring plan to better evaluate the health status of the wild boar population. Between 2011 and 2021, samples of found dead and hunted wild boar have been examined for trichinellosis, tuberculosis, brucellosis, african swine fever, classical swine fever, Aujeszky’s disease, swine vesicular disease, and swine influenza A. *Trichinella britovi* was identified in 0.001% of the examined wild boars; neither *M. bovis* nor *M. tuberculosis* were found in *M. tuberculosis* complex positive samples; 2.3% were positive for *Brucella suis*; 29.4% of the sera were positive for Aujeszky’s disease virus; and 0.9% of the samples were positive for swine influenza A virus. With an uncertain population estimate, the number of animals tested, the number of positives, and the sampling method do not allow us to make many inferences but suggest the need to implement and strengthen the existing surveillance activity, as it seems to be the only viable alternative for safeguarding animal and human health.

## 1. Introduction

Monitoring wildlife health can benefit not just animals but also human health and environmental conservation [1]. Wildlife surveillance plans have been established in many, but not all, European countries, as they are pivotal to estimating the risk of disease spillover for domestic livestock populations sharing the same living area with wild animals [2]. The economic impact of infectious diseases spreading from wildlife to livestock can be relevant, e.g., causing trade barriers on products of animal origin [3]. In the last years, interest in wild boars (*Sus scrofa*) has increased considerably, as this species is susceptible to different zoonoses and diseases affecting also domestic pigs; moreover, its population growth and adaptation to the urban and peri-urban environment represent a real direct risk also for humans from a One Health point of view. For all these reasons, wildlife surveillance plans are becoming increasingly important [4].

The number of wild boars in Italy and Europe appears to have increased significantly within the past few decades [5], even though there is no reliable estimate of their presence due to the species’ ethology [6]. The only possible estimate is based on the number of individuals killed during hunting and culling. In 2010, the Italian wild boar population was estimated at 600,000 animals [7], while, at the time of writing, researchers estimate it could have soared to about 1.5 million wild boars even if there are no publications of official national estimates [8].

The growth rates of the Italian wild boar population are similar to those of Spain, and based on hunting bags, they also appear to be in line with those of Poland, France, and Germany (between 200,000 and 640,000 wild boar per year) [5], resulting in an estimated population density throughout Europe of up to 15 individuals/km^2^ [9].

The Emilia-Romagna region is located in Northern Italy; the territory is characterized by Apennine mountains (5677 km^2^; 25.3%), hills (6202 km^2^; 27.6%), and the plain of the Po valley (10,573 km^2^; 47.7%) [10]. The Emilia-Romagna region represents about 7% of the national wild boar distribution area, corresponding to at least 60,000 wild boars potentially present in the area [11]. Wild boars are omnivores that are very adaptable to many habitat types and can quickly colonize most of them. In particular, the Tosco-Emiliano-Romagnolo Apennines turned out to be a very suitable place for the wild boar population [12]. Figure 1 shows the distribution area of wild boars in the Emilia-Romagna region.

Intraspecific interactions between wild boars and farmed pigs can easily occur, especially in outdoor pig farming. In the Emilia-Romagna region, there are 2986 farms (of which 181 are free-range farms) with 1,024,627 pigs [13]. This zootechnical sector is well developed and has great economic importance due to many high-quality foods produced in the Italian Food Valley and exported all over the world, such as “Parma ham” and other cured pork products, with a turnover of 1778 million euros in 2019 [14].

In Emilia-Romagna’s pig farms, classical swine fever (CSF) and Aujeszky’s disease (AD) are actively controlled by specific surveillance plans [15]. Until 2019, an eradication plan for swine vesicular disease (SVD) was also active in Italy [16]. All these diseases are also monitored in the wild boar population through the control of each dead wild boar showing lesions compatible with the diseases.

Wild boars have recently received increasing attention in Europe due to the introduction of ASF from the Caucasus and Russian regions [17,18,19], causing the blockage of pork products and pig exportation. Following the appearance of ASF in European territory, in 2020 a national surveillance plan was issued for african swine fever (ASF) in pigs and wild boars, which for the latter involves the search for ASF in every found dead wild boar. In 2021, the Italian ASF National Surveillance Plan also integrated the control of CSF, establishing virological control for CFS on all wild boars found dead.

The Italian national prevention plan emphasizes the importance of activating regional monitoring plans for wildlife. Unfortunately, these guidelines are not mandatory, and only a few regions have already implemented them. The Emilia-Romagna region has had its own regional monitoring plan since 2007. This is a dynamic plan, recently updated according to the national guidelines [20], and is periodically updated with regional notes on the basis of the local epidemiological situation. The Emilia-Romagna region wildlife monitoring plan includes a specific chapter dedicated to wild boars. In fact, dead animals, injured subjects in which the presence of infectious diseases is possible, and animals killed during hunting must be sampled with the help of the staff of wildlife recovery centers, the Forest Rangers, the Provincial Police, and hunters. Laboratory analyses and necroscopies are carried out by the local “Istituto Zooprofilattico Sperimentale” laboratories (IZSLER), which are part of the nationwide network of official animal health laboratories. The plan underlines that analyzing found dead animals is crucial to early detection of the introduction of pathogens in an area.

The goal of this study is to report the results of the wild boar surveillance plan in the Emilia-Romagna region over the last 11 years (2011–2021), with a focus on the most relevant diseases considered by the plan: trichinellosis, tuberculosis (TBC), brucellosis (BRC), african swine fever, classical swine fever, Aujeszky disease, swine vesicular disease, and swine influenza viruses (SwIAVs).

## 2. Materials and Methods

### 2.1. Survey Area

The study area is the Emilia-Romagna region, which is located in northeastern Italy and covers an area of about 22,510 km^2^, of which 17.5% are protected areas such as parks and nature reserves [21]. The territory is divided into nine provinces: Piacenza, Parma, Reggio Emilia, Modena, Bologna, Ferrara, Ravenna, Forlì-Cesena, and Rimini. Approximately half of the territory in the northeast is characterized by the plains of the Po valley; the remaining part further west features the Apennine Mountains and hills. Several species of wildlife are present [22]. The distribution area of the wild boar population touches all the provinces, especially in the hilly and mountainous territory, but currently we can also find these animals in the plain (Figure 1).

### 2.2. Sampling and Data Analyses

The regional monitoring plan provided for passive surveillance for all the diseases monitored, i.e., the search for compatible lesions, in:(i)all wild boars found dead at the time of the necropsy, which was systematically performed on all the carcasses conferred to the official laboratory;(ii)on hunted wild boars at the time of processing carcasses in game processing centers.

In addition, the plan established that all hunted wild boar had to be sampled for the detection of *Trichinella* spp. and provided for targeted surveillance for serological research of antibodies against CSF, AD, and SVD; virological research for ASF and SwIAv; microbiological research for TBC in hunted or dead wild boars with suspected lesion; and microbiological research for BRC in a minimum number of hunted wild boars, randomly selected at the time of carcass processing in game processing centers. The minimum number indicated by the plan was set between 60 and 100 samples per province per year (at least 540 samples for the whole Emilia Romagna region each year), assuming an expected prevalence for the diseases covered by the plan of around 5% with a C.L. of 95%.

The samples, stored at a refrigeration temperature of 4 °C, were accompanied by a form reporting the date and place of collection. For each disease, yearly descriptive statistics were provided (absolute number and percentage of positives, spatial distribution of positive samples). Wild boars hunted or found dead from 2011 to 2021 in the Emilia-Romagna region were examined as described more in detail for each single disease, in the following paragraphs.

#### 2.2.1. Trichinellosis

All wild boars found dead or shot during the hunting season were tested for *Trichinella* spp. Fifty grams of muscular tissue collected from the diaphragm was subjected to artificial digestion and submitted to microscopic examination, according to Regulation (EC) no. 1375/2015 [23] and the World Organization for Animal Health (WOAH) Manual of Diagnostic Tests and Vaccines for Terrestrial Animals [24]. The recovered larvae were preserved in 90% ethanol and sent to the International Trichinella Reference Center (ITRC) in Rome for species identification by multiplex PCR [25].

#### 2.2.2. Tuberculosis (TBC)

Organs and lymph nodes with visible lesions consistent with tuberculosis were sampled from dead wild boars during the necropsy, which is systematically performed on all the carcasses conferred to the official laboratory, and from hunted wild boars at the time of processing the carcass in game-handling centers. Specimens with lesions were subjected to histological, microbiological, and molecular analyses, according to the WOAH Manual of Diagnostic Tests and Vaccines for Terrestrial Animals [26]. Tissues were cultured for *Mycobacterium tuberculosis* in a solid medium (Lowenstein-Jensen-ST; Microbiol Diagnostici, Cagliari, Italy) and a liquid culture system (BACTEC MGIT 960, Becton, Dickinson, and Company, Franklin Lakes, NJ, USA). The samples were also tested by PCR to find *M. tuberculosis* complex (Mtbc). The isolated strains were identified by the national WOAH Reference Center (IZSLER) [27].

#### 2.2.3. Brucellosis (BRC)

At least 60–100 samples per province per year (spleen, testicles, and uterus) were randomly selected from found dead or hunted wild boars and tested for *Brucella* spp. by real-time PCR [28]. Positive samples were subjected to standard microbiological and cultural tests according to the WOAH Manual of Diagnostic Tests and Vaccines for Terrestrial Animals [29]. The isolated strains were sent to the national WOAH Reference Center (IZSAM) for species and biovar identification [29].

#### 2.2.4. African Swine Fever (ASF)

From 2011 to 2019, all dead or hunted wild boars with signs or clinical appearance leading to suspicion of swine fever were sampled and analyzed for the presence of the virus. Since 2020, as required by the Italian ASF National Surveillance Plan, all found dead wild boars, even those with no pathognomonic lesions, have been sampled, and the organs (spleen, lymph nodes, kidney, and long bone marrow) have been processed with commercial PCR kits for the detection of genomic ASFV DNA (ID Gene African Swine Fever Duplex (IDASF) kit, IDVET), according to the manufacturer’s instructions.

#### 2.2.5. Classical Swine Fever (CSF)

From 2011 to 2020, blood samples were collected from a sample of at least 60 to 100 hunted wild boars per province per year, and serological tests were performed on them. Antibodies against pestivirus (CSF-BVDV-BDV) were searched by a competitive ELISA test (kit IZS-BS), and positive samples were tested by a competitive ELISA test (kit IZS-UM) for the detection of specific CSF antibodies, according to the manufacturer directions and the WOAH Manual of Diagnostic Tests and Vaccines for Terrestrial Animals [30] in wild boar serum.

Since 2021, as required by the Italian ASF and CSF National Surveillance Plan, all found dead wild boars were sampled, and organs (spleen, lymph nodes, kidney, and long bone marrow), depending on the type of material found (carcass or remains in decomposition), were processed by a homemade PCR method for the detection of genomic CSF DNA [30,31].

#### 2.2.6. Aujeszky’s Disease (AD)

Every year, serological tests were performed on blood samples collected from at least 60–100 hunted wild boars per province. Antibodies against the AD virus’s glycoprotein E were tested using a competitive ELISA kit (Kit IZS-BS) in accordance with the WOAH Manual of Diagnostic Tests and Vaccines for Terrestrial Animals [32].

#### 2.2.7. Swine Vesicular Disease (SVD)

From 2011 to 2019, between 60 and 100 blood samples collected from hunted wild boars were serologically tested each year using the monoclonal antibody-based competitive ELISA (c-ELISA) recognized by the WOAH as the reference screening test [33]. Since 2020, research on SVD has been suspended.

#### 2.2.8. Swine Influenza A (SwIA)

From 2013, SwIAV research in wild boars’ lungs with lesions consistent with flu was implemented in the regional monitoring plan due to avian influenza outbreaks in some Emilia-Romagna poultry farms. Lung cranial lobes were sampled and homogenized 1:10 in phosphate-buffered saline (PBS). Wild boars’ lungs were analyzed in pools of a maximum of five individuals, and total RNA was extracted using the One for All vet kit (Indical Bioscience GmbH, Leipzig, Germany) following the manufacturer’s instructions. The RNAs were then screened using a TaqMan One-Step real-time RT-PCR on the influenza A virus (IAV) M gene [34].

## 3. Results

### 3.1. Trichinellosis

In the considered plan period (2011–2021), samples from 208,241 wild boars were analyzed for *Trichinella* spp., in particular: 207,710 wild boars killed while hunting and 531 wild boars found dead in the environment. The number of animals examined for *Trichinella* remained homogeneous over the years (Table 1).

Only two of the 208,241 wild boars tested positive for *Trichinella* spp.

*Trichinella britovi* was identified once in 2017 and once in 2018 (Table 1). In both cases, animals were killed during hunting and came from two municipalities in the province of Piacenza (PC) (Figure 1).

### 3.2. Tuberculosis

During the period, 317 wild boars showed suspicious tubercular lesions at necroscopy, and 71 tested positive for Mtbc PCR (Table 2). After culturing, neither *M. bovis* nor *M. tuberculosis* were found. *M. microti* was identified more frequently (19 animals), while *M. avium* was present in four cases. The positive samples were found throughout the regional territory, in seven out of nine provinces (Figure 1).

### 3.3. Brucellosis

From 2011 to 2021, samples from 8864 wild boars were tested for *Brucella* spp., with 210 positive PCR results (Table 3). Only 44 of these were positive for the cultural test, and only strains of *Brucella suis* biovar 2 were isolated. The presence of the pathogen in the wild boar population was mostly demonstrated in the provinces of Bologna, Piacenza, Parma, and Ravenna. In the province of Modena, *Brucella* spp. were detected only in 2012, and in the province of Rimini, only in 2013 (Figure 1).

### 3.4. African Swine Fever

From 2011 to 2019, passive surveillance, carried out on all 161,546 hunted wild boars and on all 251 found dead wild boars, did not identify any ASF suspected carcass. In 2020, when the Italian ASF National Surveillance Plan was implemented, 135 dead wild boars were sampled and examined, all resulting in negative results. In 2021, all 231 dead wild boars were tested, which resulted in negative results as well.

### 3.5. Classical Swine Fever

During the study decade (2011–2020), none of the 45,334 serological samples, each taken from a single wild boar, tested positive for CSF. In 2021, when the Italian ASF and CSF National Surveillance Plan was implemented, 74 samples from dead wild boar underwent PCR tests for CSF, all scoring negative.

### 3.6. Swine Vesicular Disease

Between 2011 and 2019, 36,083 individual serological samples taken from as many wild boars were negative when tested for SVD.

### 3.7. Aujeszky’s Disease

Over the eleven years of the plan, 45,331 sera collected from hunted wild boars were analyzed for AD, and 13,497 resulted in positive results, with an average positivity of 29.4% (Figure 2). The seroprevalence appears to have been increasing since 2015, with no differences at the provincial level.

### 3.8. Swine Influenza

From 2013 to 2021, 4054 lung samples were collected from hunted or found-dead wild boars. These samples were tested by the SwIAV one-step real-time RT-PCR reactions, and 37 of them resulted in positive results (Table 4).

## 4. Discussion

Climate and land exploitation changes have had a significant impact on wildlife population dynamics [35,36]. In particular, land abandonment in mountainous and hilly territories has been very marked in Northern Italy during the last decades, influencing the expansion of the range of wild ungulates and their population size [37].

The increase in wild boar populations draws attention to the variety of infectious and parasitic diseases these animals may harbor as natural hosts [38]. The monitoring plan carried out in the Emilia-Romagna region over the last eleven years has made it possible to examine more than 208,000 wild boars, testing for diseases that are important for both animal and human health. The number of wild boars sampled every year depends on the number of wild boar carcasses found in the regional provinces and on the number of wild boars hunted as indicated in the wildlife hunting plan. In fact, the majority of wild boar samples come from the provinces of Parma and Bologna, where more consistent population control plans have been active [11].

Due to the lack of systematic sampling and the opportunistic nature of our data collection (which characterizes the structure of passive surveillance), we cannot draw any inferences about the regional wild boar population’s health status. Passive surveillance based on found dead wild boars relies on the personal efforts of Forest Rangers, the Provincial Police, and hunters and depends on their consciousness and sensitivity relative to wildlife diseases. Consequently, only 251 dead wild boars were retrieved and delivered to the official laboratories from 2011 to 2019, with an average of 27 carcasses per year. Since 2020, following the expansion of ASF in Europe, the search for carcasses has been strongly recommended, leading to the recovery of six times more wild boars compared to all the previous years.

*Trichinella* spp. is a food-borne zoonotic agent. The main risk for humans is the consumption of raw or undercooked Suidae or equid meat. In Europe, there are only four circulating species (*T*. *britovi*, *T*. *spiralis*, *T*. *nativa*, and *T*. *pseudospiralis*). According to European regulation [23], wildlife surveillance for the presence of *Trichinella* spp. has the aim of assessing the risk of contamination of farmed pig carcasses, so animals of the susceptible species must be tested. Foxes and other carnivores are considered *Trichinella* spp. territorial indicators. In Italy, *T*. *britovi* is endemic to several wild species [39], and it was found in wild boars in the Emilia-Romagna region once in 2017 and once in 2018 (0.005%), but only in the province of Piacenza. These results are lower than those reported by other studies in southern Italy, in which they observed a prevalence of 0.01% of *Trichinella britovi* in wild boars [40] and other European countries (e.g., Croatia [41] and France [42]). In the south-east Balkan countries (i.e., Bulgaria, Romania, and Serbia [43]), Poland, and Spain, the rates of *Trichinella* spp. infections each year in wild boar [44] are higher. In the world panorama, a high prevalence of *Trichinella* in wild boar has been detected in countries such as Argentina with 25% [45] and Iran with 5.7% of wild boars positive for *Trichinella britovi* [46]. In 2010, *T. pseudospiralis* was isolated for the first time from a wild boar in the Emilia-Romagna region [47]. In 2017, *T*. *spiralis*—the most dangerous species for humans—was found for the first time in a dead fox in the province of Piacenza [48]. In the USA, *Trichinella spiralis* larvae were identified in wild boar meat samples [49]. Mixed infection by *T. spiralis* and *T. britovi* in a wild boar was reported in Spain [50] and mixed infection by *T. spiralis* and *T. pseudospiralis* in wild boars in Germany [51]. The monitoring plan revealed the sporadic presence of *Trichinella britovi* in the wild boar population in two municipalities in the Piacenza province (Figure 1). Although the results of monitoring more than 208,000 wild boars within the last 11 years seem not alarming, it is nevertheless important to sensitize hunters to have all hunted animals checked before consumption for food biosafety purposes.

Tuberculosis, caused by the *Mycobacterium tuberculosis* complex (MTBC), is a widespread zoonotic disease in domestic and wild animals. Its presence in wildlife often goes unnoticed due to the absence of symptoms. The wild boar is susceptible to *M. bovis*, and it is regarded as an important host in European wildlife [52], together with the Eurasian badger *Meles meles* in Great Britain and Ireland and deer belonging to the subfamily Cervinae in several European regions [53]. The detected genotypes are the same as those found in domestic animals, although their prevalences in wild populations and domestic ones may differ [54]. Our study showed the presence of wild boars with granulomatous lesions in organs and lymph nodes compatible with mycobacteriosis. In fact, *Mycobacterium* spp. was isolated from 23 of the 71 PCR-positive samples, and in most cases, *M. microti* was identified. The isolation of *M. microti* is quite frequent, and its health impact on humans and domestic animals is minimal or nonexistent [55]. Recent studies in southern Italy highlighted a moderate presence of *M. bovis* in the wild boar population, which is probably related to domestic outbreaks [56]. The highest prevalence of *M. bovis* in wild boar can be observed in the Iberian Peninsula, where 30% of wild boars die naturally from TB [57]. Uncontrolled circulation of the infection in wildlife could jeopardize the eradication of the disease, therefore changing the territory’s health status recognized by the European Community.

Our data over an eleven-year period do not suggest a significant role of wild boars as reservoirs for bovine tuberculosis in the Emilia-Romagna region, which has been declared officially free from bovine tuberculosis since 2007 [58].

The genus Brucella includes nine species, of which *B. abortus*, *B. melitensis*, and *B. suis* are the agents of bovine, small ruminant, and swine brucellosis, respectively. These bacteria cause economic losses and important morbidity in domestic and wild animals and have been frequently isolated in wildlife. There are five different biovars of *B. suis*, of which biovars 1, 2, and 3 cause infection in domestic and feral pigs. Biovars 1 and 3 are the most frequently detected and are zoonotic agents. Biovar 2 is an emerging problem in wildlife as it causes reproductive disorders, including infertility, and could represent a serious health risk for domestic pigs [59]. *Brucella* spp. seroprevalence varies widely in different European countries; *B. suis* is mainly present in wild boars and in European brown hares (*Lepus europaeus*) [60,61,62,63,64]. A global comprehensive literature review and meta-analysis of *Brucella* in pigs from 2000 to 2020 reported that the overall prevalence of brucellosis in wild boars between 2006 and 2010 was 22.3%, while after 2010, the prevalence gradually decreased after the WOAH proposed control safety standards for animal production [65]. Our study, directly researching the etiological agent, confirmed a low circulation of the disease among the wild boar population in the Emilia-Romagna region, as reported in other Italian regions [66,67]. The positivity found is caused by *Brucella suis* biovar 2, the most isolated in European wild boars. In pigs, it often causes infertility and/or abortion, potentially associated with miliary lesions in the reproductive system, while in wild boars, it generally does not involve obvious lesions and may not cause any symptoms [68]. Spillover infection from wild boars to cattle has been previously reported in Europe, but the pathogenicity of *B. suis* biovar 2 in cattle is unknown. In fact, the infection sometimes occurs subclinically but still results in a positive reaction to routine brucellosis tests in the context of cattle eradication campaigns [69]. As some of the aspecific positive reactions to serological tests for brucellosis in officially disease-free territories could be explained by environmental contamination, the role of wild animals in the circulation of the infection should be further investigated.

CSF affects the family Suidae and is caused by a Pestivirus belonging to the family Flaviviridae. Therefore, wild boars are susceptible to the virus, as are domestic pigs, and depending on their population density, they could become a dangerous reservoir, leading to significant economic losses in domestic pig farming. Since the presence of this highly contagious disease in wild species constitutes a great economic and health risk for the domestic pig population and the pig industry, CSF is listed among notifiable diseases [70]. Control of CSF in wild boars relies primarily on passive surveillance, and hunters play a key role in collecting samples. Once a suspected outbreak is confirmed, specific territorial measures are applied, which provide for the definition of the area at risk and the surveillance area around it [71]. Its diffusion is currently limited to some areas of Central and Eastern Europe [72,73], while it is endemic in large parts of the world, including Latin America, Eastern Europe, and Asia [74]. As expected, no circulation of the virus was detected based on the epidemiological situation in Italy. In fact, Italy is CSF free, as are the other EU states, North America, Australia, and New Zealand [75].

ASF is a devastating disease for the pig industry and is gradually spreading in Europe [76]. ASF is endemic in sub-Saharan regions of the African continent. In Europe, the disease has been endemic since 1978 on the island of Sardinia (Italy), and in 2007, the disease made its first appearance in the Caucasus regions (i.e., Georgia, Armenia, and Azerbaijan) and subsequently in Russia, Ukraine, and Belarus [77]. In 2014, the virus reached the countries of the European Union [78] (i.e., Lithuania, the Baltic States, and Poland [79]). In 2017, the infection also affected the Czech Republic and Romania. Belgium, Hungary, and Bulgaria were infected in 2018, while Slovakia and Serbia were infected in 2019 [80]. The worst possible scenario occurred in 2018 when ASFV was detected in China, which contains half of the world’s swine population [81]. Widespread dissemination in China has been followed by spread to Mongolia, Vietnam, Cambodia, North Korea, South Korea, Myanmar, Laos, and the Philippines [82]. The last European country, in chronological order, to be affected by the current epidemic wave is Italy, where the first case of ASF was confirmed in January 2022 [83]. North Macedonia and Thailand also reported the first appearance of the disease in January 2022, and in March 2022, ASF was first reported in Nepal [84]. The alarm and concern are determined by the fact that, if ASF enters the pig industry, it would be necessary to proceed with the culling of all heads to eradicate the virus, whose characteristics include strong environmental resistance [85]. Our study found no positive samples in the wild boars we examined (year 2011–2019), but at the time of writing, ASF outbreaks have been recorded in three Italian regions: Piedmont, Liguria, and Lazio, with an estimated R_0_ of 1.41 in Piedmont/Liguria and 1.66 in Lazio [83]. The Italian outbreaks seem limited, and according to Loi et al. [83] and Salazar et al., 2022 [86], the strong impact of the fragmentation of territories (i.e., sites into urban areas and the linkage of these sites via roads and railroads) has limited the propagation of ASF, contrary to a homogeneous landscape less anthropized, which seems to favor its diffusion, e.g., in Polonia, where in 2019 there were 2477 cases diagnosed of ASF [87] or in Lithuania with the maximum prevalence of 87.5% in May 2018 [88]. Surveillance of found-dead wild boars should allow the early detection of the virus, but for the passive surveillance system to be effective, a large number of carcasses must be found and analyzed. In 2021, the Italian ASF National Plan was amended, introducing a minimum number of wild boars to be tested for passive surveillance per each region, amounting to 270 animals for the Emilia-Romagna region. In a population of at least 60,000 wild boars, the recovery of only 231 dead animals does not appear to be sufficient to ensure the surveillance system’s sensitivity. For this reason, we encourage the adoption of ecological sampling schemes for detecting wild boar carcasses in the environment [89]. We also recommend that members of local communities and outdoor recreationists (e.g., hunters, anglers, and mushroom pickers) be encouraged, through tailored communication campaigns [90], to report wild boar carcasses in the wild.

SVD is an infectious disease caused by an Enterovirus belonging to the family Picornaviridae. Although this disease is considered moderately contagious, it can cause significant economic losses in pig farming. Therefore, it is important to evaluate its circulation in wild boar populations. In Italy, a national plan was established to achieve eradication in pig farms [91], according to the European Council Directive 92/119/EEC [92], and the status of a country free from SVD was reached in 2019. In the Emilia-Romagna region, no SVD cases have been detected on domestic pig farms since 2006 [16]. At present, the virus is endemic in several countries in South and Central America, Asia, parts of Eastern Europe, and neighboring countries, while the presence of the virus in Africa is still unknown [93]. The data collected from 2011 to 2019 confirm the absence of this disease in the wild boar populations of the Emilia-Romagna region, after which monitoring of wild boars was suspended.

AD, or pseudorabies, is caused by Suid herpesvirus type I, family Herpesviridae, subfamily Alphaherpesviridae. AD is an enzootic disease in feral pigs and wild boars, and the latter appear to be the main maintenance hosts. Members of the family Suidae are the natural hosts, but the virus can also infect many other mammals. The wide circulation of the disease in wild boars makes it necessary to study the potential epidemiological role of the species. However, some studies highlighted that wild and domestic populations seem to maintain two distinct strains, implying that the risk of a re-emergence from wild to domestic species is low [72]. The prevalence of wild boars appears to be dynamic and changes every year for unknown reasons [94]. In Emilia-Romagna, in parallel with the wild boar population growth, the AD seroprevalence has started to slightly increase since 2015 (Figure 2). In pig farms, the same type of investigation showed a progressive reduction of AD prevalence from 20% to 0% over the last eleven years [95], demonstrating that vaccination and biosecurity measures are successful in preventing the entry into farms of the virus present in wild boars. In Italy, a marked difference was observed between strains isolated in wild boar or hunting dogs and strains isolated in working dogs on pig farms, suggesting the presence of two different infection cycles and two distinct ecological niches [96]. Apparently, spillovers between wild boars and pigs are not a frequent occurrence [97,98]. In this study, we observed a seroprevalence of 29.4%, showing a strong synergy with what was found in the European panorama [99,100,101]. However, AD viral circulation necessitates systematic checks to prevent possible impairment of the eradication plan in the regional territory.

SwIAVs are RNA viruses belonging to the Orthomyxoviridae family. Wild waterfowls are natural hosts, while wild boars and domestic pigs can become spillover [102], and they have a pivotal role in the development of reassortant strains [103]. Furthermore, wild boars and wild birds share the same ecological niche, thus enabling the possibility of IAV spillover events between these species [104]. Although the IAV circulation in swine hosts is dynamically investigated worldwide, the IAV prevalence in wild boars and feral pigs is poorly researched [104]. Finally, IAVs might be considered a potential public health threat since these viruses are frequently respiratory pathogens for each mammal species, and the occurrence of zoonotic and, even more often, reverse zoonotic transmission of IAVs is very high. Therefore, knowledge of viral hypervariability is at the basis of One Health epidemiological surveillance, both in domestic and wildlife hosts [102,103]. In Europe, antibodies belonging to three subtypes of IAVs—H1N1, H1N2, and H3N2—were found in wild boars, and their prevalence appears to vary greatly depending on the country and region [72]. Serological studies performed in several European countries, such as Poland, Spain, Slovenia, and Germany, emphasized the strong variability of SwIAV seroprevalence, which varies from 0% (e.g., Slovenia) to 26% (e.g., Southern Germany) depending on the region [104]. The role of wild boars in the ecology of influenza has not been clarified yet, but monitoring allows evaluating the possible maintenance and diffusion in both wild and domestic populations [105,106]. The data analyzed since 2013 in the frame of the Emilia-Romagna surveillance plan has shown an apparently low prevalence of infection in the wild boar population (about 1% over the years), in line with other European studies that have already reported a low prevalence or total absence of SwIAV elsewhere [104].

## 5. Conclusions

The constant expansion of the wild boar population could act as an important transmission route for different diseases. In this scenario, monitoring plays a key role, as the growth of wild animal populations leads to increased attention as well as research and the detection of disease cases. The results obtained over the last eleven years within the Emilia-Romagna region monitoring plan suggest the need to maintain and improve the health surveillance of the wild boar population to obtain increasingly representative data. Despite the strategy of monitoring and controlling wild populations being recommended by the WOAH [107] as one of the most useful activities for safeguarding global animal and human health, monitoring alone is not sufficient to reduce the circulation of pathogens in wild boars or prevent the possible spread of disease in farmed pigs. On the other hand, in Italy, wildlife is considered an “unavailable heritage” of the state; it is protected, and the killing of sick or infected animals is unlawful except in specific cases that must be identified in agreement with the European Commission. Therefore, knowing in which areas a disease is present in wild boars can be decisive for implementing greater biosecurity measures on farms and especially at the farm entrance (thorough cleaning and disinfection of buildings, transport vehicles, and personal protective equipment) in order to minimize as much as possible the risk of a disease spillover from wild boars to the domestic pig population.

In this One Health perspective, the structure of the wildlife monitoring program in the Emilia-Romagna region could be further extended to include other emerging diseases.

## Figures and Tables

**Figure 1 animals-13-01832-f001:**
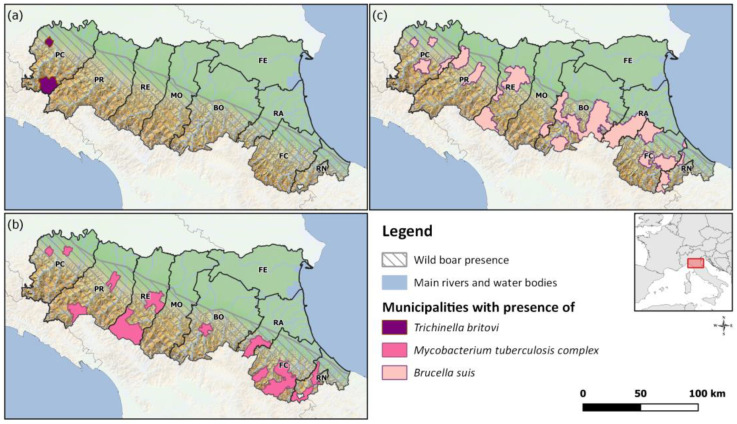
Distribution area of wild boars in the Emilia-Romagna region: (**a**) municipalities where the presence of *Trichinella britovi* has been detected (2011–2021); (**b**) municipalities where the presence of *Mycobacterium tuberculosis* complex has been demonstrated (2011–2021); (**c**) municipalities where *Brucella suis* has been isolated (2011–2021).

**Figure 2 animals-13-01832-f002:**
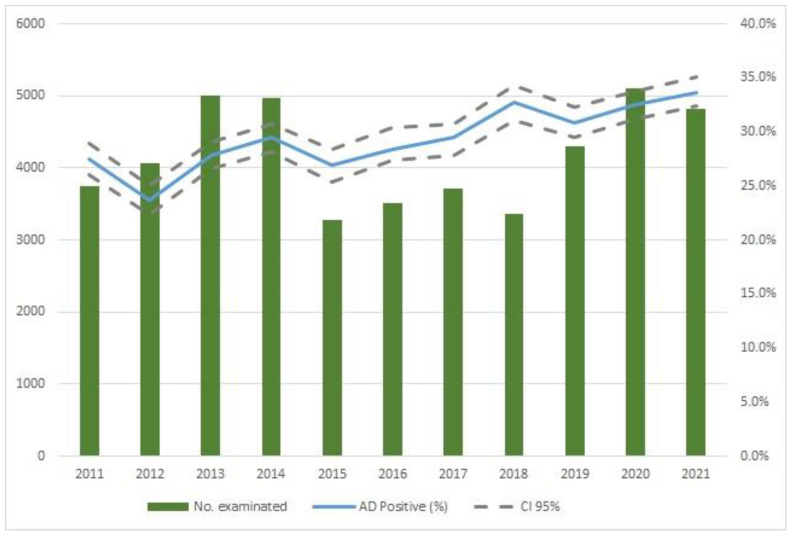
Number of wild boars examined per year and trend in the percentage of AD seroprevalence from 2011 to 2021.

**Table 1 animals-13-01832-t001:** Number of wild boars examined that tested positive for *Trichinella* spp. per year in the Emilia-Romagna region from 2011 to 2021.

Year	No. of Animals Examined	No. of *Trichinella* spp.-Positive Sample
2011	15,499	0
2012	16,147	0
2013	15,434	0
2014	15,614	0
2015	17,589	0
2016	19,756	0
2017	20,732	1
2018	19,475	1
2019	21,329	0
2020	19,830	0
2021	26,836	0
Total	208,241	2

**Table 2 animals-13-01832-t002:** Samples tested for the *M. tuberculosis* complex and numbers of typed samples per year from 2011 to 2021.

Year	No. of Animals Examined	No. of Mtbc PCR- Positive Samples	*M. microti*	*M. avium*
2011	19	8	5	0
2012	1	0	0	0
2013	17	11	0	0
2014	17	6	3	0
2015	27	10	4	1
2016	59	10	3	1
2017	41	2	0	0
2018	41	11	1	2
2019	7	4	2	0
2020	43	6	1	0
2021	45	3	0	0
Total	317	71	19	4

**Table 3 animals-13-01832-t003:** Number of wild boars tested for *Brucella* spp. per year from 2011 to 2021.

	No. of Samples Examined	No. of *Brucella* spp. PCR-Positive Samples	No. of *Brucella* spp. Microbiological-Positive Samples *
2011	1121	35	4
2012	833	20	1
2013	1019	36	6
2014	1028	14	4
2015	846	13	4
2016	522	15	5
2017	401	14	8
2018	480	19	5
2019	631	12	1
2020	964	14	4
2021	1019	18	2
Total	8864	210	44

* *Brucella suis* biovar 2.

**Table 4 animals-13-01832-t004:** Samples of wild boar’s lungs tested per year for Swine influenza in the Emilia-Romagna region from 2013 to 2021.

	No. of Samples Examined	No. of PCR-Positive Samples
2013	720	2
2014	511	4
2015	306	1
2016	203	0
2017	181	0
2018	242	2
2019	421	3
2020	718	23
2021	752	2
Total	4054	37

## Data Availability

The data were obtained from the annual reports of the Emilia Romagna wildlife monitoring plan, available for online consultation at https://archive.izsler.it/pls/izs_bs/v3_s2ew_consultazione.mostra_pagina?id_pagina=2652, accessed on 14 May 2023.

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
