# Peer review of "Eleven Years of Health Monitoring in Wild Boars (Sus scrofa) in the Emilia-Romagna Region (Italy)"

_animals, 2023, doi:10.3390/ani13111832_

Round 1

Reviewer 1 Report (Previous Reviewer 2)

The authors carried out and considered the recommendations of the reviewers, resolving them satisfactorily.

Only minor comments remain regarding specific situations, highlighted in the attached .pdf file.

Author Response

We thank the referee for his/her comments and advices. 

Reviewer 2 Report (Previous Reviewer 1)

The authors made the necessary corrections, but in some cases I have doubts, such as Line 311 - 27 wild boars a year until 2019, what kind of test is this for such a large region additionally divided into provinces. It would have to be described in a different way (this period). In some places, the authors downplayed my comments - they have the right to do so, but I still believe that my comments would improve this manuscript. I still share the view that the discussion is too extensive. I leave these comments to the Editors, and the manuscript is recommended for publication after any further minor corrections at the request of the Editors.

Author Response

We thank the referee for his/her comments and advices.

Lines 310-317: the period has been rewritten to accommodate the reviewer’s request.

About the lengthy discussion, we expanded this section following the request for "expanded bibliography" and the "inclusion specific comparison to other regions of Italy or Europe in the case of specific diseases." requested by the referee in his/her previous report.

This manuscript is a resubmission of an earlier submission. The following is a list of the peer review reports and author responses from that submission.

Round 1

Reviewer 1 Report

It should be said that the authors undertook quite interesting research, which may be of both application and practical importance, especially since they concern wild boars, which remain the basic reservoir and vector of ASFV. In addition, wild boars are an interesting species because in recent years the reproductive potential of their population has increased significantly, and thus there has been a rapid increase in numbers and local densities. As the authors themselves emphasize, there is a lack of reliable data and reliable methods for assessing the size of the population and its dynamics in individual years. Using data only from hunting is subject to error, as it depends on many factors related to the attractiveness of hunting and local and national attitudes to hunting, and thus hunting pressure on this species. Therefore, the data presented in the study, despite the fact that they refer to the size of specific samples, do not necessarily reflect the actual epizootic threat to the wild boar population, and in some cases, the epidemiological threat.

After reading and analyzing the study, some comments came to my mind, which do not detract from the results obtained and the research carried out, but they should be clarified:

There is no precision in the methodology from how many wild boars the samples actually came from?

In a few cases, it is mentioned that samples were taken from 60-100 hunted wild boars, in others there is no such information. Were these studies homogeneous? What was the key to collecting these samples, after all, it is said earlier that there are about 60,000 wild boars in the region. Very low percentage of samples in relation to the population?

Line 133-135, what were the changes to qualify the animal for sampling?

Line 149 -150, tests for the presence of ASF were carried out only in wild boars shot with clinical symptoms. After all, an animal can be a reservoir of the virus without clinical symptoms (!) With swine flu, there is no information at all from how many animals the samples came from?

Only this information can be found in the results. In my opinion, they should be included in the methodology and it is necessary to add information why so many samples and how do they represent the state of the population?

Line 231-232, how many carcasses were there?

Why have swine flu samples been collected since 2013, and not as in the methodology?

Line 269-271, in the discussion Authors mention that the number of samples depends on the size of the population, I cannot find a sampling algorithm in their studies. Please clarify what were the abundance estimates and sampling key?

Line 274-276, why the opportunistic nature of sampling?

Line 279-281, redundant information generally known.

Line 293, there were only two species of Trichinella, so the wording "different" seems to be invalid?

Line 296-297, abundance is not the only reason the parasite reservoir increases or decreases?

Line 300-301, according to Figure No. 1, Trichinella was detected only in one part of the study region, and thus is it for sure in the entire region, unjustified inference?

Line 330-331, there is nothing in the methodology about biotype identification?

In the discussion, the authors devote a lot of space to epidemiological issues, while in fact it does not follow directly either from the title or from the set goal of the work? It needs to be clarified somehow. In addition, the discussion should be a kind of comparison of my own results with the results of other authors on the same issue or issues, but in this study I feel a lack of real discussion. In the discussion, I also feel the lack of a specific comparison to other regions of Italy or Europe in the case of specific diseases. Are the reported cases (relative to the size of the population) many or few?

Line 418-422, in this suggestion it is not clear what supervision is meant, whether monitoring of the population (improved) or monitoring of zoonotic risk in the wild boar population?

Line 422-425, will the harmonization of sample collection improve the epizootic situation and reduce the epidemiological risk?

The conclusions should be reconstructed so that they concern the indication of specific solutions, based on the results obtained, the mere increase in monitoring will definitely improve the state of knowledge, but it will not cure the situation of zoonoses and the resulting threat.

The literature could be expanded especially with the occurrence of ASFV or trichinellosis in other regions or countries, e.g. .2020.25.24.1900527,

DOI: http://dx.doi.org/10.15666/aeer/1805_68416856, DOI

10.21521/mw.6018, these are only some suggestions, but it is worth referring to the situation in other places. In addition, it is worth mentioning the dynamics of wild boar numbers in other countries.

Although all the cases mentioned do not significantly diminish the very concept of the study, but may affect the conclusions obtained and the assumed implementation of the study's goal, therefore, in my opinion, they require revision and refinement.

After a thorough explanation of the issues described by me and the correction of the text in the indicated fragments, I recommend the study for printing.

Reviewer 2 Report

The aims of this study was to report the results of 11 years of the wild boar surveillance plan in the Emilia-Romagna region in Italy, with focus on the most relevant diseases.

Specific comments/suggestions are highlighted in the attached .pdf file.

In general terms, the manuscript is of great importance, considering the new One Health approach that is related to zoonotic diseases that involve wild boar in their transmission. Considering the interaction between wild boar, farming pigs and humans. It is important to check for the name of the pathogens that should be in italic, also check for the estructure of tables and figures.  
